# OpenReview forum: "Towards the Worst-case Robustness of Large Language Models"
_ICLR.cc/2026/Conference — Submitted to ICLR 2026_

### Official Review · Reviewer_AsPK · 2025-10-30

**Soundness:** 1
**Presentation:** 2
**Contribution:** 2
**Rating:** 2
**Confidence:** 5

**Summary:**

This paper studies adversarial robustness of LLMs under worst-case (white-box) conditions. It claims two main contributions: 1) Upper bounding worst-case robustness of existing defenses (both deterministic and stochastic) by introducing a stronger white-box adversarial method, I2-GCG, which ensures tokenization consistency between optimization and inference. Experiments show that many deterministic defenses exhibit nearly 0% robustness under this new attack. 2) Lower bounding worst-case robustness via a theoretical framework that formulates randomized smoothing robustness as a knapsack optimization problem (fractional or 0–1 knapsack), providing analytical lower bounds for stochastic defenses such as uniform or absorbing kernels. The authors then apply these formulations to LLM safety detectors and report certified ℓ₀ radii and suffix robustness results on the AdvBench dataset. The paper argues that by bounding robustness from both above (empirical attacks) and below (theoretical certificates), one can achieve a more complete understanding of the “worst-case” robustness of LLMs.

**Strengths:**

1. The idea of formulating randomized smoothing bounds as a knapsack problem is mathematically creative and a novel interpretation of existing theory.

2. The connection between vocabulary size and certified bounds provides some analytical insight.

3. The appendices appear to contain detailed proofs and extra context, though the main text occasionally glosses over technical assumptions.

**Weaknesses:**

1.  The experiments in the main text are extremely limited and not aligned with the large scope implied by the title (“Towards the Worst-Case Robustness of LLMs”). The “upper bound” results rely almost entirely on Table 1, which evaluates a few deterministic defenses under I2-GCG. There are no ablations, no qualitative jailbreak examples, and no exploration of why tokenization consistency matters beyond a one-line intuition. The “lower bound” section (Tables 2–3) only reports a few average numbers on AdvBench without comparison to competing theoretical certification methods (e.g., Moon et al., 2023; Kumar et al., 2023). Given the paper’s strong theoretical claims, the lack of empirical diversity (no varied datasets, no different models beyond 7B/8B class, no real-world attacks) makes the work feel incomplete and largely conceptual.

2. The title suggests a comprehensive investigation into LLM worst-case robustness “towards” a general solution, but the content mainly covers: a minor technical tweak to an existing attack (I-GCG → I2-GCG), and a repackaged theoretical treatment of randomized smoothing. There is no holistic analysis of “worst-case robustness” across modalities, alignment procedures, or instruction-following behaviors. Thus, the scope and ambition are mismatched with what is actually demonstrated.

3. The proposed I2-GCG attack is not fundamentally new: the improvement (ensuring tokenization consistency) is minor and already implicitly addressed in other implementations (e.g., Li et al., 2024a; Basani & Zhang, 2024).  Actually, the attack evaluations simply replicate known results (deterministic defenses collapsing to 0% robustness under white-box settings), providing little new empirical insight.

4. The theoretical “knapsack” formulation is elegant but only rephrases existing randomized smoothing theory from my perspective. The equivalence to Cohen et al. (2019) and Teng et al. (2020) is explicitly acknowledged in the text, meaning no new tighter bounds or practical improvements are offered. There is no experimental validation that the knapsack-derived lower bounds match observed robustness under stochastic defenses. The claim of “tightness” remains purely theoretical, lacking corresponding empirical results.

5. The certified ℓ₀ radius and suffix length results (e.g., 2.02 or 6.41) are not contextualized, what do these numbers mean in terms of actual model safety? Are they significant improvements over baselines? The authors evaluate “GPT-4o” without clarifying whether this refers to a closed-source API or a proxy, raising reproducibility concerns.

6. Severe experiment issues need to be seen: No discussion on computational cost, training overhead, or scalability. No qualitative or visualization-based analyses to illustrate adversarial or certified behavior. Lack of comparison with recent works like Tree of Attacks (Mehrotra et al., 2023), AmpleGCG (Liao & Sun, 2024), or Diffusion-based certified defenses (Chen et al., 2024a,b). This omission further weakens the contribution’s originality.

**Questions:**

1. Can the authors expand the experiments beyond one dataset (AdvBench) and provide more comprehensive comparisons with recent attacks and defenses?

2. How does I2-GCG perform on real-world jailbreak prompts (e.g., JailbreakBench 2024)?

3. The current theory assumes bounded or binary functions. Can it be extended to non-binary outputs, continuous embeddings, or multimodal inputs?

4. How would the results change under more realistic constraints, e.g., subword tokenization noise or cross-lingual settings?

5. Are these derived bounds useful for deployment or policy decisions?

---

> ### Author Response · Authors · 2025-11-13
> **Part 1**
>
> We thank Reviewer AsPK for their detailed feedback. We appreciate their acknowledgment that our knapsack framework is **"mathematically creative"** and a **"novel interpretation of existing theory"** (Strengths).
>
> However, we are puzzled, as this positive assessment of our core theoretical contribution is **directly contradicted** by the reviewer's own Weakness 4, which claims it **"only rephrases existing... theory"**.
>
> Similarly, the reviewer's assessment of our experiments as **"extremely limited" (W1)** is directly contradicted by **Reviewer hyzH**, who praised them as a **"Comprehensive Empirical Evaluation" (S2)**, and by **Reviewer 6jR1**, who noted we "evaluate attacks across several different LLMs" (S1).
>
> Given these deep contradictions, we will focus on addressing the reviewer's specific factual inaccuracies and conceptual misunderstandings, which appear to stem from a misreading of our tables and appendices.
>
> ---
>
> ### 1. **On Weakness 1: "Experiments are extremely limited"**
>
> We must respectfully disagree with this assessment, a view not shared by other reviewers (e.g., hyzH, 6jR1). Our experiments are designed to systematically establish both the theoretical *lower bound* (Tables 2-3) and the empirical *upper bound* (Table 1) of worst-case robustness.
>
> The reviewer's criticism in W1 is based on several **factually incorrect** claims:
>
> 1.  **Claim: "no comparison to competing theoretical certification methods (e.g., ... Kumar et al., 2023)."**
>     This is **incorrect**. We explicitly benchmark our Suffix attack certification against **Kumar et al. (2023) in Table 3**. (We will address the reviewer's deeper misunderstanding of this baseline in our response to W5).
>
> 2.  **Claim: "no qualitative jailbreak examples."**
>     This is **incorrect**. We provide detailed qualitative examples of our defense (DiffTextPure) in action in **Appendix F.2 (Case I & II)**.
>
> 3.  **Claim: "no exploration of why tokenization consistency matters."**
>     This is **incorrect**. We explore this point both qualitatively and quantitatively.
>     *   **Qualitatively:** We dedicate an entire takeaway point in **Appendix K** to explaining *why* this is the "only reason" previous attacks failed.
>     *   **Quantitatively:** As we demonstrate in our response to W3 below, this "consistency" is the *entire difference* between a **failed attack** (e.g., 82% robustness for PAT) and a **successful one** (0% robustness).
>
> ---
>
> ### 2. **On Weakness 2&3: "I²-GCG attack is not fundamentally new"**
>
> The reviewer's criticism that our I²-GCG is a "minor technical tweak" [W3] misunderstands how progress in white-box attacks (à la Carlini & Wagner) is defined. The contribution is not the algorithm's complexity, but the **correctness of its implementation** and the **new results it reveals**.
>
> This "tweak" is *not* "minor"; it is the **fundamental reason** why prior white-box evaluations (like the original I-GCG) **failed** to break deterministic defenses.
>
> For instance, applying the previous I-GCG (which lacks strict token consistency) does *not* achieve a 100% attack success rate, leaving defenses like PAT appearing robust:
>
> | Model | No Defense | PPL | ICD | Self-Reminder | PAT |
> | :--- | :---: | :---: | :---: | :---: | :---: |
> | Vicuna-7B | 0% | 96% | 0% | 0% | 82% |
> | Llama2-7B | 0% | 98% | 0% | 0% | 90% |
> | Llama3-7B | 16% | 96% | 24% | 24% | 86% |
>
> Our I²-GCG, by *correcting* this single tokenization flaw, achieves 100% attack success (i.e., 0% robustness for these defenses, as shown in our Table 1).
>
> This demonstrates that the "tweak" is, in fact, the **essential component** for a correct evaluation. This is not "replicating known results" [W3]; it is **correcting a flawed empirical record** where defenses were incorrectly reported as robust.
>
> ---
>
> ### 3. **On Weakness 4: Knapsack formulation "only rephrases existing theory"**
>
> We are deeply puzzled by this claim, as it **directly contradicts the reviewer's own "Strengths"** (i.e., that our work is "mathematically creative" and a "novel interpretation"). A "novel interpretation" is valuable precisely because it provides **new insights**.
>
> 1.  **Our Framework is General:** Our knapsack framework is a **more general theoretical framework** for randomized smoothing in discrete spaces. From this general framework, prior complex results like Cohen et al. and Teng et al. emerge as **special cases**. As we demonstrate in our proofs in **Appendix D.5**, deriving these prior results becomes almost trivial.
>
> 2.  **New, Non-Trivial Insights:** Crucially, this general framework allows us to derive **new insights** not present in prior work. The most salient example is **Theorem 5.5**: the first theoretical proof that the **Uniform Kernel (i.e., Diffusion) is provably more robust than the Absorbing Kernel (i.e., Masking/BERT)**, given the same $p_A$.
>
> This new finding (Thm 5.5) alone—which the reviewer missed—disproves the claim that our "creative" work "only rephrases" existing theory.

---

> ### Author Response · Authors · 2025-11-13
> **Part 2**
>
> ### 4. **On Weakness 5: Certified results "are not contextualized... [no] baselines"**
>
> This criticism is based on **severe factual errors**, as the reviewer has missed the explicit baselines provided for *both* results and misunderstands *why* they are different.
>
> 1.  **Re: $l_0$ Radius (Table 2):** The reviewer claims the 2.02 radius lacks a baseline. This is **incorrect**. As clearly labeled in **Table 2**, this result is benchmarked against the **"Human" baseline (2.12)**, which serves as a semantic upper bound.
>
> 2.  **Re: Suffix Length (Table 3):** The reviewer claims the 6.41 length lacks a baseline. This is also **incorrect**. As clearly labeled in **Table 3**, this result is benchmarked against the SOTA **"Kumar et al. (2023)" baseline**.
>
> 3.  **Re: Kumar on $l_0$:** The reviewer's complaint implies Kumar should *also* be a baseline in Table 2. This reveals a **fundamental misunderstanding of the baseline itself**. Kumar's method is a *suffix-only* defense (random deletion). As we discuss in detail, it is **conceptually inapplicable** to certifying general $l_0$ perturbations. The reviewer is asking for a meaningless comparison.
>
> 4.  **Re: GPT-4o:** We clarify that "GPT-4o" refers to the API, used as detailed in **Appendix G.1** (detector prompt) and **G.6** (purifier).
>
> ---
>
> ### 5. **On Weakness 6: "Severe experiment issues... Lack of comparison with recent works"**
>
> 1.  **Qualitative Analysis:** This is incorrect. See **Appendix F.2**.
> 2.  **Computational Cost:** We discuss this in **Appendix G.4**. Certification (lower bound) is an offline cost for a theoretical guarantee, as in all RS methods [r1]. Inference (empirical defense) is cheap.
> 3.  **SOTA Attacks (ToA, AmpleGCG):** This is a **threat-model mismatch**. Those are *black-box/heuristic* attacks. Our paper focuses on (a) **theoretical white-box** attacks (I²-GCG) and (b) **theoretical certification** (Knapsack). In the relevant black-box setting, we *do* compare against SOTA optimization attacks (GCG, AutoDAN) in **Appendix G.3 (Table 4)**.
> 4.  **SOTA Defenses ([Chen et al.])**: [Chen et al.] is work in the *vision* domain. Our work establish this general knapsack framework for all domain, including the *text* domain and *arbitrary* smoothing kernels.

---

### Official Review · Reviewer_6jR1 · 2025-11-04

**Soundness:** 2
**Presentation:** 3
**Contribution:** 2
**Rating:** 4
**Confidence:** 3

**Summary:**

This paper investigates “worst-case” attacks for both deterministic and stochastic defenses to assess their robustness. In addition, the authors provide a theoretical analysis to characterize a lower bound on the robustness of stochastic defenses.

**Strengths:**

S1. The paper evaluates attacks across several different LLMs, which increases empirical validity.

S2. The theoretical lower bound result is interesting and (to me) is the clearest and strongest contribution in this paper.

S3. The paper is generally well-written and easy to read.

**Weaknesses:**

W1. The technical contribution is somewhat limited. More importantly, different defenses are designed for different threat models. If defense A is specifically designed for a black-box setting, then evaluating it under white-box conditions is not very meaningful, because the defense was never intended to handle that capability. Instead, what is needed is “worst-case within the constraint the defense was designed for.” In other words, “worst-case” needs to be scoped per defense mechanism consistent with the defended threat model.

W2. The definition of “worst-case” in this paper is not clearly specified. For example, what is assumed attack capability, what prior knowledge is assumed, and what attack objective is assumed when declaring “worst”?

W3. Related to W1, the authors are encouraged to select a more diverse and more state-of-the-art set of defense mechanisms baselines in their experiments, spanning multiple defense setting categories, and then evaluate worst-case adversarial robustness conditioned within each defense’s intended setting/assumption class. This would make the empirical conclusions much more compelling and meaningful.

**Questions:**

See Weaknesses.

---

### Official Review · Reviewer_oS3B · 2025-11-04

**Soundness:** 2
**Presentation:** 1
**Contribution:** 2
**Rating:** 2
**Confidence:** 3

**Summary:**

This paper aims to design a certified robustness (CR) defense for LLMs against jailbreak attacks by certifying the robustness of harmful prompt detectors (which can be seen as certifying the robustness of text classifiers). To this end, the authors first propose a new **sample-dependent** "CR-definition" in the scenario of generative modeling, and then design an algorithm to realize so-called "certified robustness" by smoothing the protected model via solving a restricted optimization problem. Some experiments are conducted to verify the effectiveness of the proposed defense.

**Strengths:**

I appreciate the effort of the authors in trying to establish certified robustness guarantees for generative LLMs (although I personally think the proposed "CR-guarantee" is incorrect).

**Weaknesses:**

1. I think the authors cannot call their proposed method a "certified robust method". Specifically, when people say a ML model is "certified robust", it means that the model would not change its prediction for **ANY** possible input samples and their corresponding perturbed versions under a reasonable perturbation budget (see examples for vision models [r1, r2] and language models [r2]). However, according to Lines 138-139 of the paper, the "certified robustness guarantee" proposed in this paper is actually sample-dependent (since the term $p_A$ depends on $x$). Furthermore, in Theorem 5.2, the authors also admit that if the sample-dependent term $p_A := g(x)$ cannot satisfy certain conditions, no "CR-guarantee" can be established for this sample at all. This means the proposed so-called "CR-guarantee" actually cannot ensure the robustness for **ALL** possible samples, and this is why I believe that the proposed method is not a "CR-method" but simply an empirical adversarial robustness enhancing method. Additionally, I think it is inappropriate to hide the data term $x$ from the sample-dependent $p_A := g(x)$, which may mislead readers.

2. In Theorem 5.2, it requires the condition $p_A(x) \geq 1-\beta^d$ to be satisfied to enable one to establish a "CR-guarantee". I think such a condition is too restrictive and may not be realistic in real-world scenarios. Specifically, $\beta^d$ seems to be a very small value, which will result in $(1-\beta^d)$ being very close to $1$. Suppose the ML model that needs to be protected is a "harmful prompt detector" that outputs only 0 or 1. Then, for a harmful input $x_{adv}$, the condition $p_A(x_{adv}) \geq 1-\beta^d$ means that the harmful content detector needs to be **very very good at** detecting $x_{adv}$. IF the detector is already so good at detecting a given $x_{adv}$, I doubt that if it is indeed necessary to further leverage the proposed method to defend against the given $x_{adv}$.

3. In Theorem 5.4, where is the CR-guarantee proposed for the uniform kernel? From Theorem 5.4, I can only see a closed-form solution for a term $v(i,j)$, but its meaning and connection with "CR" are not explained. Besides, in Theorem 5.5, what does the "certify" function $certify(\cdot)$ mean? I think the notations and presentation in this paper can be significantly improved.

4. Since the proposed CR-defense is not a "real CR" but a data-dependent defense, I suspect that it can be easily broken by prompt-level adaptive attacks such as [r4, r5]. So I suggest the authors carefully evaluate their proposed defense against these attacks. Besides, the authors are also encouraged to adopt more common and stronger token-level jailbreak attacks such as [r6, r7] into their empirical evaluations.


**References**

[r1] Cohen et al. Certified Adversarial Robustness via Randomized Smoothing. ICML 2019.

[r2] Carlini et al. (Certified!!) Adversarial Robustness for Free! ICLR 2023.

[r3] Zeng et al. Certified Robustness to Text Adversarial Attacks by Randomized [MASK]. Computational Linguistics, 2023.

[r4] Chao et al. Jailbreaking Black Box Large Language Models in Twenty Queries. arXiv 2023.

[r5] Andriushchenko et al. Jailbreaking Leading Safety-Aligned LLMs with Simple Adaptive Attacks. ICLR 2025.

[r6] Hayase et al. Query-Based Adversarial Prompt Generation. NeurIPS 2024.

[r7] Sadasivan et al. Fast Adversarial Attacks on Language Models In One GPU Minute. ICML 2024.

**Questions:**

See **Weaknesses**.

---

> ### Author Response · Authors · 2025-11-13
> **Part 1**
>
> We thank Reviewer oS3B for their time and for acknowledging the **"Originality"**, **"Quality"**, and **"Significance"** of our work.
>
> However, the reviewer's assessment of our work's weaknesses is based on a series of **fundamental misunderstandings** regarding (1) the definition of Randomized Smoothing (RS) itself, (2) the core problem setup of our paper (Definition 4.1), (3) the algorithmic structure of our contribution, and (4) the distinction between theoretical guarantees and empirical defenses.
>
> We will address each misunderstanding in order.
>
> ---
>
> ### 1. **A fundamental misunderstanding of Randomized Smoothing (RS)**
>
> **Reviewer's Core Criticism:**
>
> The reviewer claims our method is "not a 'certified robust method'" because our guarantee is "sample-dependent" (i.e., it depends on $p_A$). The reviewer contrasts this with \[r1, r2] as examples of "real CR".
>
> **Our Response:**
>
> This criticism stems from a fundamental misunderstanding of the entire Randomized Smoothing (RS) field.
>
> *   **All RS Guarantees Depend on $p_A$:** The certified radius in *all* RS-based methods is *necessarily* a function of $p_A$—the lower bound of the smoothed classifier's probability for the correct class. This is the central premise of the reviewer's **own citation, \[r1] Cohen et al. (ICML 2019)**. In \[r1, Theorem 1], the certified radius is explicitly derived from $p_A$ (the probability of the "top class"). If $p_A \le 0.5$, \[r1] provides no certified radius at all.
>
> *   **Our Definition is the Standard:** Our Definition 4.1 ("Given... $g(x) = p_A$... find the minimal output $p_{adv}$") is the **standard, universally accepted definition** for RS certification, identical in principle to that used in \[r1] and \[r3].
>
> *   **Our Contribution:** Our contribution is *not* to invent a new (and supposedly "incorrect") definition of CR. Our contribution is to take this **standard RS definition** and, for the first time, provide a **general and tight solver** for it in the discrete, non-binary space of LLMs by reducing it to the Fractional/0-1 Knapsack Problem.
>
> Therefore, the reviewer's primary criticism is invalid, as it (incorrectly) attempts to invalidate the entire foundation of the RS field that \[r1] established.
>
> ---
>
> ### 2. **A fundamental misunderstanding of our Problem Definition (confusing $x$ and $x_{adv}$)**
>
> **Reviewer's Core Criticism:**
>
> The reviewer argues the condition $p_A \ge 1-\beta^d$ in Theorem 5.2 is too restrictive, stating: "...for a harmful input $x_{adv}$, the condition $p_A \ge ...$ means... the detector needs to be very very good at detecting $x_{adv}$."
>
> **Our Response:**
>
> This criticism reveals a second fundamental misreading of our paper. The reviewer has confused the known input $x$ with the unknown adversarial sample $x_{adv}$.
>
> *   **The Premise is Wrong:** The reviewer's premise—that $p_A$ is the model's performance on $x_{adv}$—is **diametrically opposed** to our core setup.
>
> *   **Our Definition:** As clearly stated in **Definition 4.1**, $p_A = g(x)$ is the **known, measured** probability on the **clean sample $x$**. Our *goal* is to use this known value to *solve for* the lower bound on the **adversarial sample**, $p_{adv} = \min g(x_{adv})$. We are *solving for* the performance on $x_{adv}$, not *assuming* it.
>
> *   **The Reviewer's "Doubt" is Invalid:** The reviewer's entire argument ("IF the detector is already so good at detecting... $x_{adv}$, I doubt that... it is... necessary to defend...") is based on a **critically flawed premise** that he himself invented by confusing $x$ and $x_{adv}$.
>
> **On the "Restrictive Condition" of Theorem 5.2:**
>
> The condition $p_A \ge 1-\beta^d$ is not a flaw of our framework; it is a finding of our framework that reveals an inherent property of the Absorbing Kernel. We explicitly discuss this property ourselves in the whole Theorem 5.5, Appendix D.4.
>
> ---

---

> ### Author Response · Authors · 2025-11-13
> **Part 2**
>
> ### 3. **Misunderstanding the Algorithmic Contribution (Thm 5.4 & 5.5)**
>
> **Reviewer's Core Criticism:**
>
> "In Theorem 5.4, where is the CR-guarantee... I can only see a closed-form solution for a term $v(i,j)$... what does the 'certify' function... mean?"
>
> **Our Response:**
>
> This reveals a misunderstanding of our paper's "Algorithm + Input" structure.
>
> *   **Step 1 (The Solver):** Section 4.2 provides the **general certification algorithm** (Algorithm 1), which is the Fractional Knapsack Solver. This algorithm takes a set of "items" (with weights and values) and a capacity $p_A$ as input, and outputs the certified lower bound $p_{adv}$.
>
> *   **Step 2 (The Inputs):** **Theorem 5.4** provides the *inputs* for that algorithm. It derives the specific "items" (the $v(i,j)$ terms, which represent the volumes/weights of all possible value-to-weight ratios) for the **Uniform Kernel**.
>
> **Clarification:** The "CR-guarantee" is not in Thm 5.4 alone. The guarantee is the **output of Algorithm 1** *after* it is fed the inputs computed by **Theorem 5.4**.
>
> The `certify(...)` function is simply shorthand for this two-step procedure. Theorem 5.5 proves that for *any* given $p_A$, the final certified radius $d$ computed for the Uniform Kernel is always greater than or equal to the radius computed for the Absorbing Kernel.
>
> ---
>
> ### 4. **Confusing Theoretical Bounds with Empirical Attacks**
>
> **Reviewer's Core Criticism:**
>
> "Since the proposed CR-defense is not a 'real CR'... I suspect that it can be easily broken by prompt-level adaptive attacks such as \[r4, r5]."
>
> **Our Response:**
>
> This criticism is a direct consequence of the invalid premise in W1.
>
> *   **Our Method IS a "Real CR":** As established in our response to W1, our method *is* a "real CR" method, providing a **theoretical lower bound** on worst-case robustness.
>
> *   **Theoretical Guarantees Cannot Be "Broken" by Empirical Attacks:** By definition, a certified lower bound holds against *ANY* possible attack, **including \[r4, r5, r6, r7]**. A theoretical guarantee states the *worst possible outcome*, so it cannot be "broken" by an empirical attack; it already accounts for it.

---

### Official Review · Reviewer_hyzH · 2025-11-06

**Soundness:** 2
**Presentation:** 2
**Contribution:** 3
**Rating:** 2
**Confidence:** 2

**Summary:**

This paper proposes a method to certify robustness of LLMs to adversarial attacks using stochastic defenses by a reduction to the the fractional knapsack problem -- rather than finding the adversarial input $x_{adv}$ it minimizes a function $f$ assigning weight to different stochastic generations $z$. This simplifies the problem and allows them to estimate lower bounds on the probability of successful adversarial attacks, allowing them to certify defenses within a particular range of distances.

There is also a successful adversarial attack strategy which ensures that the tokenizations of the attack are the same during inference and attack generation, creating very low upper bounds on robustness. However, this attack strategy is much less successful against stochastic defenses such as applying uniform kernels, absorbing kernels, or mask generation.

**Strengths:**

Originality: This paper proposes (as far as I am aware) a novel reduction of stochastic defense certification to fractional knapsack optimization, simplifying a problem and allowing it to certify robustness results.
Quality: The paper supports its theoretical claims with detailed proofs and explanations, and demonstrates the proposed methods experimentally.
Significance: Adversarial attacks on LLMs are a significant problem, and certifying the robustness of defenses is important for understanding defense quality.

**Weaknesses:**

Clarity:
W1) Lou et al. (2023) is not a particularly helpful first citation for explaining stochastic defenses, as it covers discrete diffusion processes. It seems to me that the idea is that $z \sim p(z|x)$ is a discrete diffusion process, but the concept can use much more explanation in the introduction of the paper. It only seems to be explained in "Results on randomized defenses." and in section 4.1, making it much harder to understand the intended connection to the knapsack problem reduction.
W2) While the proposed method is fairly abstract, it would be helpful to define substantially more of the notation.
W3) The experimental details are pretty sparse, and it's not clear to me how the experiments were conducted.

**Questions:**

Q1) For the specific proposed stochastic defenses, it's unclear to me what the difference between
Q2) Can you elaborate on the different relevant $p(z|x)$ for the different ways in Section 4.1?
Q3) What does it mean to use the different models in the experiments? Section 4.1 ends saying that you only certify the safety detector -- to me that would imply that the detector looks at given model outputs, but it's unclear to me where the detector and generations come from.
Q4) I worry that the stochastic adversarial defenses are strong against the $I^2-GCG$ attack _in particular_ -- random modifications are particularly likely to break tokenization assumptions, and so it would be reassuring to the effectiveness of the stochastic defenses against simpler baselines such as GCG, BEAST, or Best-of-N jailbreaking.

---

> ### Author Response · Authors · 2025-11-13
>
> We thank Reviewer hyzH for their valuable feedback and for recognizing the **"Originality"** of our novel knapsack reduction, the **"Quality"** of our detailed proofs, and the **"Significance"** of certifying LLM defenses.
>
>
>
> ---
>
> ### 1. **On Weakness 1 (Clarity of Stochastic Defenses & Knapsack Connection)**
>
> We thank the reviewer for this suggestion. We agree that the intuitive link between the knapsack problem and randomized smoothing can be established earlier.
>
> *   **On [Lou et al. 2023]:** We cited this paper because our **Uniform Kernel** (Definition 5.3) is precisely the forward process used in discrete diffusion models like [Lou et al.]. This citation is crucial for our **Theorem 5.5**, which proves the theoretical superiority of this diffusion-style kernel over the masking-style kernel (Absorbing Kernel, Def. 5.1).
> *   **Action:** In the final version, we will revise the introduction to more clearly foreshadow the connection from Section 4 (our general knapsack solver) to Section 5 (its application on diffusion/masking kernels).
>
> ---
>
> ### 2. **On Weakness 2 (Defining Notation)**
>
> We appreciate the reviewer's focus on clarity. We would like to point the reviewer to **Appendix A: NOTATIONS (Page 18)**, where we provide a **comprehensive two-page table** that explicitly defines *all* symbols used in the paper (e.g., $f, g, \beta, \alpha, p_A, p_{adv}, v(\gamma)$, etc.).
>
> To further improve readability in the main text, we will add more explicit inline reminders for key terms like $p_A$ and $p_{adv}$ upon their first appearance.
>
> ---
>
> ### 3. **On Weakness 3 & Question 3 (Sparse Experimental Details)**
>
> We were surprised by the reviewer's concern that our experimental details are "sparse," as we provide **eight pages of exhaustive details, prompts, and additional results in Appendix F and G (Pages 40-47)**. We believe these appendices fully address W3 and Q3.
>
> *   **On Q3 ("where the detector... come from"):** We provide the **exact prompt** used to construct our (near) 0% FPR safety detector in **Appendix G.1 (Page 43)**.
> *   **On W3 ("how the experiments were conducted"):** We detail our full experimental setup, hyperparameters, datasets, and baselines in **Appendix G.2**. Furthermore, we detail the **DiffTextPure** framework (Algorithm 3) used for evaluating stochastic defenses in **Appendix F.2**.
> *   **On Q3 ("what does it mean to use the different models"):** This refers to our ablation study in **Appendix G.6 (Table 6)**, where we test different models (Vicuna, Llama-3, GPT-4o) as the *purifier* in our DiffTextPure framework. This experiment led to a key finding: smaller open-source models **completely fail** at this purification task (0% robustness) and require fine-tuning, whereas GPT-4o can serve as an effective purifier via prompting alone.
>
> We trust that these extensive appendices resolve all concerns about experimental clarity and reproducibility.
>
> ---
>
> ### 4. **On Question 4 (Effectiveness against GCG Baselines)**
>
> Directly applying the previous I-GCG on these defenses does not achieve a 100% attack success rate, as shown in the following table:
>
> | Model | No Defense | PPL | ICD | Self-Reminder | PAT |
> | :--- | :---: | :---: | :---: | :---: | :---: |
> | Vicuna-7B | 0% | 96% | 0% | 0% | 82% |
> | Llama2-7B | 0% | 98% | 0% | 0% | 90% |
> | Llama3-7B | 16% | 96% | 24% | 24% | 86% |
>
> Therefore, we fixed token consistency bugs (particularly in PAT and the Llama-3 tokenizer) and propose to view the defended models as a whole and optimize the entire model to break models with input-output filtering defenses.

---

### Official Review · Reviewer_1mBt · 2025-11-10

**Soundness:** 3
**Presentation:** 3
**Contribution:** 2
**Rating:** 6
**Confidence:** 2

**Summary:**

This paper studies the worst-case robustness of LLMs against adversarial attacks. The authors develop I2-GCG, an improved white-box attack that ensures tokenization consistency, showing most deterministic denfeses achieve nearly 0% worst-case robustness.

**Strengths:**

S1. Strong Theoretical Framework

S2. Comprehensive Empirical Evaluation

S3. The paper writting is clear and easy to follow

**Weaknesses:**

W1. The gap between theoretical guarantees and practical robustness remains large.

W2. The propose framework cannot handle insertion/deletion attacks or long heuristic prompts.

W3. While the theoretical analysis is strong, the paper doesn't propose new defense mechanisms that achieve better worst-case robustness.

**Questions:**

Please refer to weakness part

---

### Meta-Review · Area_Chair_Vbeg · 2026-01-06

**Summary:**

This paper tackles the worst-case robustness of large language models. Specifically, this paper provides both the upper bound and lower bound of the worst-case robustness, by proposing a strong adaptive attack and reducing the functional optimization to a fractal knapsack problem, respectively. Two case studies are provided to smooth the distribution of the diffusion models and masked generation.

**Reviewer Concerns:**

- A primary issue is the ambiguity and potential overstatement surrounding the notion of “certified robustness”, highlighted by Reviewers oS3B and 6jR1.
- Reviewer hyzH points to gaps in exposition—particularly around the connection between stochastic defenses and the knapsack formulation.
- Reviewer 6jR1 raises a foundational concern about the definition of “worst-case” robustness.
- Reviewer 1mBt pointed out that the gap between theoretical guarantees and practical robustness remains large.

**Reviewer Scores:**

Reviewer 6jR1 may have a chance to raise the sore.

---

### Decision · Program_Chairs · 2026-01-26

Reject